# SIFM: A Foundation Model for Multi-granularity Arctic Sea Ice Forecasting

## Abstract

Arctic sea ice performs a vital role in global climate and has paramount impacts on both polar ecosystems and coastal communities. In the last few years, multiple deep learning based pan-Arctic sea ice concentration (SIC) forecasting methods have emerged and showcased superior performance over physics-based dynamical models. However, previous methods forecast SIC at a fixed temporal granularity, e.g. sub-seasonal or seasonal, thus only leveraging inter-granularity information and overlooking the plentiful inter-granularity correlations. SIC at various temporal granularities exhibits cumulative effects and are naturally consistent, with short-term fluctuations potentially impacting long-term trends and long-term trends provides effective hints for facilitating short-term forecasts in Arctic sea ice. Therefore, in this study, we propose to cultivate temporal multi-granularity that naturally derived from Arctic sea ice reanalysis data and provide a unified perspective for modeling SIC via our **S**ea **I**ce **F**oundation **M**odel. SIFM is delicately designed to leverage both intra-granularity and inter-granularity information for capturing granularity-consistent representations that promote forecasting skills. Our extensive experiments show that SIFM outperforms off-the-shelf deep learning models for their specific temporal granularity.

## 1 INTRODUCTION

Arctic sea ice has a profound influence on both local and global climate systems. The near-surface air temperature of Arctic regions has increased at a speed that is two to nearly four times faster than the global average from 1979 to 2021, a phenomenon known as "Arctic amplification" (Screen & Simmonds, 2010; Rantanen et al., 2022). This accelerated temperature rise performs a key role in the unprecedented rapid diminishing of Arctic sea ice which has extensive consequences that could transcend the polar area. For example, the accelerated reduction of Arctic sea ice could not only jeopardize the survival of species residing in polar regions but also pose adverse effects on local communities whose livelihoods and well-being depend on those animals; it could substantially affect mid-latitude summer weather by weakening the storm tracks (Vavrus, 2018); and it will bring new opportunities for marine transportation and new access to natural resources like fossil fuels (Vincent, 2020).

Due to its vital role in coastal communities, global climate, and potential impacts on the world's economy, numerical and statistical models have been proposed to forecast pan-Arctic sea ice concentration (SIC) ranging from sub-seasonal to seasonal scale (Johnson et al., 2019; Wang et al., 2019). However, numerical and statistical models usually rely on high-performance computing on CPU clusters and often lead to complex debugging processes and uncertain parameterization, which limits their performance in forecasting long-term SIC changes. With the advent of deep learning models and their powerful capability in capturing complex patterns within high dimensional data, recent studies have developed end-to-end SIC forecasting models based on deep learning approaches and have presented a promising performance that exceeds previous numerical and statistical methods (Andersson et al., 2021; Ren et al., 2022). Although the intrinsic annual cyclic trend and intra-seasonal predictability of Arctic sea ice (Wang et al., 2016) contains rich information both between and within temporal scales, existing deep learning-based methods mainly focus on predicting SIC at a specific temporal granularity, e.g., 7 days or 6 months' averages, which leads to potential neglect of intrinsic correlation between different time scales and limits the performance of forecasting models. Since the Arctic sea ice extent (SIE, where SIC value is larger than 15%) has been ob-

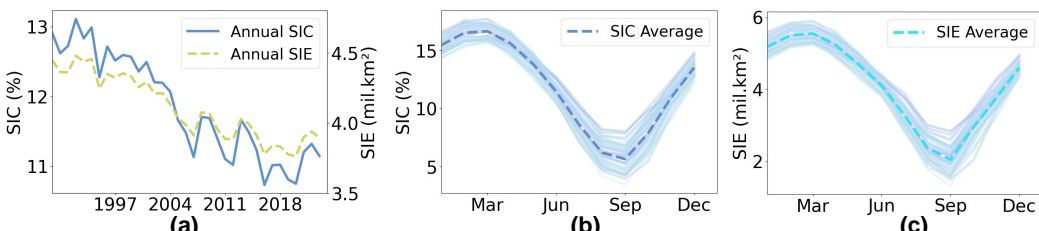

Figure 1: **Visualization of Arctic sea ice trends.** (a)The annual average SIC and SIE trend over the last 35 years (1987-2023); the monthly cyclic trend of SIC **(b)** and SIE **(c)**. Note that the averaged SIC values are calculated over the entire pan-Arctic region which could only be used to observe the trend.

served a continuous reclining trend during the last few decades (Figure 1(a)) and a clear recurrent variational pattern, i.e., the annual pan-Arctic sea ice edge usually starts to expand after the summer melting season in September (Figure 1(b)), utilizing inter-granularity and intra-granularity information could be mutually beneficial. For instance, long-term trends in weekly granularity could help to calibrate short-term daily predictions and finer granularity features could provide more accurate initial conditions to facilitate seasonal forecasting. Besides, the most commonly utilized U-Net architecture (Ronneberger et al., 2015) in previous work (Andersson et al., 2021) implicitly fulfills sequential modeling by channel-wise fusion operations. The prediction of future SIC is essentially a spatio-temporal forecasting task involving the prediction of over a hundred thousand time series, each representing a non-overlapping grid location in the Arctic region. We argue that considering forecasting future SIC is obviously a spatio-temporal task, and explicit modeling of SIC sequences could improve forecasting skills.

Based on the above-mentioned motivations, we propose the transformer-based **S**ea **I**ce **F**oundation **M**odel (**SIFM**) that unifies the temporal granularity of interest to boost overall performance on forecasting SIC in pan-Arctic region. Unlike previous approaches (as demonstrated in Figure 2), we propose to independently tokenize spatial features, explicitly extract sequential information and jointly model three granularities: daily, weekly average, and monthly average. Specifically, SIFM first embeds SIC from each temporal granularity into independent spatial tokens and sequentially concatenated to represent temporal fluctuations within each granularity. Then, we treat these independent sequences as correlated granularity variates and utilize the attention mechanism in conjunction with the feed-forward network (FFN) for extracting both intra-granularity and inter-granularity correlations. By incorporating multi-granularity representation, SIFM could simultaneously generate future SIC in different temporal scales and boost overall performance. Our contributions are three folds:

- We revisit the potentially overlooked inter-granularity information by previous methods for Arctic SIC forecasting and explore the effectiveness of independent spatial tokens representation of SIC for facilitating accurate predictions.

- We propose SIFM that leverages independent spatial tokenization of SIC and effectively unifies three temporal granularities that cover from sub-seasonal to seasonal scale for better overall representation and improved forecasting performance.

- The comprehensive experiments demonstrate that by adopting the approach of multi-granularity fusion, our SIFM achieves state-of-the-art on prediction in each granularity, which advances toward a more practical Arctic sea ice forecasting system.

## 2 RELATED WORKS

### 2.1 SEA ICE CONCENTRATION FORECASTING

Researchers have proposed various approaches to forecasting SIC, encompassing numerical and statistical models (Wang et al., 2013; Yuan et al., 2016). However, numerical and statistical models

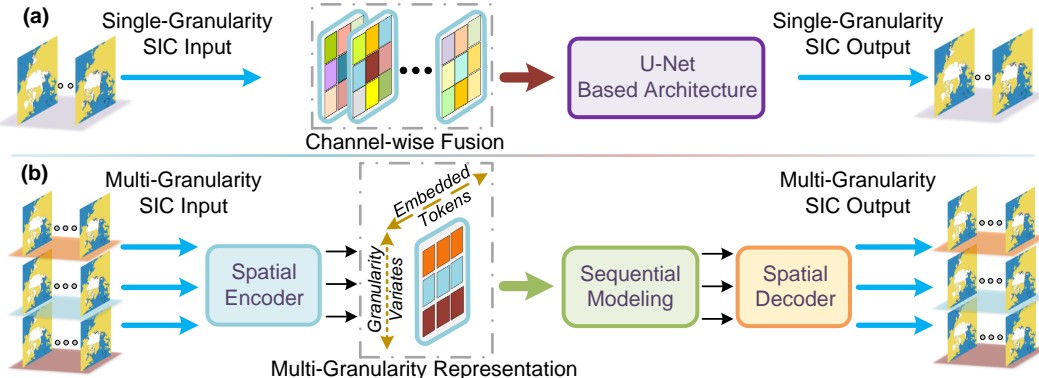

Figure 2: **The main differences** between **(a)** existing mainstream SIC forecasting approaches and **(b)** our SIFM are follows: (1) Previous models take a channel-wise fusion to jointly extract spatial features, e.g., utilizing 2D convolution to expand and downsample SIC channels. In our case, we focus on capturing effective spatial tokens representation of SIC by the shared spatial encoder. (2) The correlation among input SIC sequence is implicitly modeled via the U-Net-based architecture in **(a)** while SIFM explicitly captures intra-granularity and inter-granularity correlation via sequential modeling. (3) We propose leveraging multi-granularity information that is naturally derived from the SIC and embedding it into granularity variates to improve overall forecasting skills.

usually rely on the high-performance computing of the CPU cluster and tend to result in complex debugging processes and uncertain parameterization. Recently, deep learning models have drawn the attention of sea ice research communities and have been widely investigated for Arctic sea ice forecasting (Petrou & Tian, 2019; Kim et al., 2020; Ali et al., 2021; Ali & Wang, 2022). These methods utilize U-Net-based architectures to solve daily (SICNet (Ren et al., 2022), or monthly (IceNet (Andersson et al., 2021), MT-IceNet (Ali & Wang, 2022)) SIC forecasting. However, although these U-Net-based architectures are built on top of LSTM (Liu et al., 2021) or CNN (Andersson et al., 2021), the temporal information inherent in sea ice modeling can not be fully exploited. Moreover, these methods and the latest Transformer-based model (Zheng et al., 2024) concentrate on single-granularity SIC forecasting, where the inter-granularity information from multi-granularity sea ice modeling is overlooked.

## 2.2 MULTI-SCALE REPRESENTATIVE LEARNING

The multi-scale phenomenon is common in vision tasks, while it is always overlooked in sea ice modeling. To exploit the information in multi-scale sources, multi-scale features are commonly exploited by using spatial pyramids (Lazebnik et al., 2006), dense sampling of windows (Yan et al., 2012), and the combination of them (Felzenszwalb et al., 2008) in the vision community. The learning of CNN-based multi-scale representations is typically approached in two ways: utilizing external factors like multi-scale kernel architectures and multi-scale input architectures (Reininghaus et al., 2015), or designing internal network layers with skip and dense connections (Lin et al., 2016). Recently, there has been a surge of interest in applying transformer-based architectures to computer vision tasks, with the Vision Transformer (ViT) being particularly successful in balancing global and local features compared to CNNs (Dosovitskiy, 2020). When revisiting the task of forecasting sea ice concentration, its multiscale features stem from different temporal resolutions. Existing methods focus on a single scale, such as daily, weekly, or monthly. However, different temporal resolutions are inherently connected, and treating them as a single scale for modeling would increase the complexity of network learning.

## 3 SIFM FOR MULTI-GRANULARITY ARCTIC SEA ICE FORECASTING

Given historical Arctic SIC records $Y = \{X_{T-L-1}, ..., X_{T-1}, X_T\} \in [0\%, 100\%]^{L \times H \times W}$, where $L$ is the input length of a specific granularity which includes the given observation time step $T$, $H$ and $W$ denotes the rectangle pan-Arctic region, the forecasting model predicts the subsequent

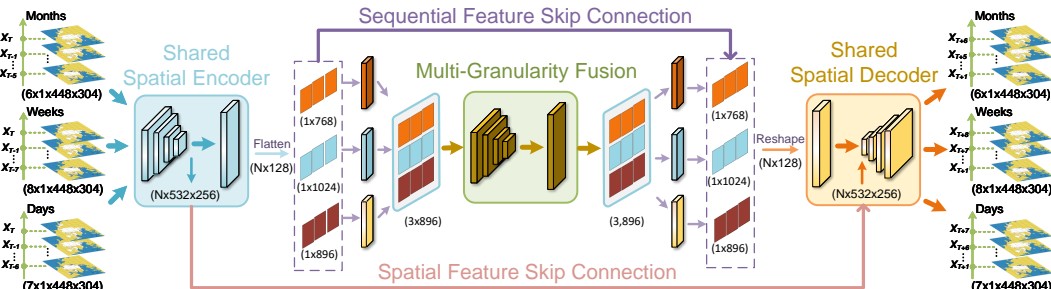

Figure 3: **Overview of proposed SIFM**, which comprises three main components: **(1)** The **shared spatial encoder** first independently extracts spatial features of input SIC from each granularity (i.e. 7 days, 8 weeks' averages and 6 months' averages) to obtain spatial tokens, and then concatenates these spatial tokens accordingly. **(2)** The embedded spatial tokens are subsequently flattened with respect to their granularity and linearly projected into the same length. We propose to utilize an encoder-only transformer backbone to perform **multi-granularity fusion** which explicitly captures both inter-granularity and intra-granularity sequential features. **(3)** Lastly, the predicted multi-granularity features are restored to the shape of the input via linear transformation and the **shared spatial decoder**.

SIC values $\hat{Y} = \{X_{T+1}, ..., X_{T+P-1}, X_{T+P}\} \in [0\%, 100\%]^{P \times H \times W}$ with forecasting lead times of $P$. In this study, our SIFM jointly models three granularities, i.e., daily, weekly average, and monthly average SIC values that cover both sub-seasonal and seasonal variations, and simultaneously forecasting on all these temporal scales. For each temporal granularity, the input length $L$ equals the forecasting lead times $P$. The overview of the proposed SIFM architecture is presented in Figure 3. The shared spatial encoder and decoder perform SIC tokenization and restoration while multi-granularity fusion explicitly extracts sequential information.

### 3.1 SEA ICE CONCENTRATION TOKENIZATION

Existing mainstream deep learning-based methods for SIC forecasting adopt U-Net architectures and leverage 2D convolution to perform channel-wise expansion and downsampling that extracts both spatial features and temporal dependencies. However, since U-Net-based models are not specifically designed for sequence modeling (Azad et al., 2024), the joint spatial-channel fusion of SIC and implicit sequence modeling could be ill-posed properties for spatio-temporal forecasting tasks. In this regard, we propose to independently tokenize spatial features at first, which could disentangle the above ill-posed problem and be beneficial for SIC forecasting.

**Independent spatial embedding.** Since we aim to simultaneously model SIC derived from three temporal granularities, encoding their spatial features into shared embedding space not only yields consistent representation but also reduces the number of trainable parameters. Inspired by prior works (Hu et al., 2023; Chen et al., 2023), we utilize Swin Transformer V2 (Liu et al., 2022) as the backbone for both shared spatial encoder and decoder.

Specifically, each SIC input is independently fed into the shared spatial encoder and partitioned by a non-overlapped window to generate patch representation (Dosovitskiy, 2020) with 32 spatial channels (the original SIC data has only one channel). To preserve local SIC information, we choose the smallest 2 by 2 window size for the patch partition. Then, the patch tokens are further transformed by the first two Swin Transformer blocks. The multi-scale spatial features are extracted through the subsequent hierarchical encoder layers which comprise a patch merging operation and two Swin Transformer blocks. The patch merging operation first concatenates the spatial feature of each group of 2 by 2 adjacent patch representations from the previous encoder layer. The calculation of each

pair of two consecutive Swin Transformer blocks in encoder layers can be described as follows:

$$z_s^b = LN(WMSA(z^{b-1})) + z^{b-1},$$
$$z^b = LN(MLP(z_s^b)) + z_s^b,$$
$$z_s^{b+1} = LN(SWMSA(z^b)) + z^b,$$
$$z^{b+1} = LN(MLP(z_s^{b+1})) + z_s^b, \tag{1}$$

where $z_s^b$ and $z^b$ represents the output spatial feature of the (**S**hifted) **W**indow-**M**ulti-head **S**elf **A**ttention module and the MLP module for block $b$, respectively; LN denotes the layer normalization operation (Lei Ba et al., 2016). The attention mechanism with a shifted window could effectively extract neighboring SIC information and sufficiently represent the local correlation of sea ice. After all input SIC are independently encoded into 2D spatial features, we apply linear projection to generate 1D token for each SIC to obtain compact spatial representation for sequential modeling.

The shared spatial decoder adopts an identical Swin Transformer backbone and the decoding procedure is symmetrical to the encoding process, except that the patch merging operation is replaced by the patch expanding operation (Cao et al., 2022). While patch merging downsamples the input spatial feature dimension and increases the embedding channels, patch expanding symmetrically restores the resolution of the feature map and merges channels via linear transformation.

**Spatial feature skip connection.** Since the SIC features encoded by Swin Transformer blocks will be tokenized into highly compact sequence representation, the spatial SIC information should be maximally preserved during the sequential modeling. Besides, our proposed sequential modeling backbone comprises deep encoding layers which might lead to loss of embedded spatial features. To preserve spatial SIC information and avoid insufficient restoration, we propose to add a skip connection between the output of the last pair of Swin Transformer blocks in the spatial encoder and the input of the first block in the shared decoder (see in Figure 3).

## 3.2 Multi-granularity fusion

We propose to jointly model three granularities that cover sub-seasonal to seasonal scale, i.e., 7 days, 8 weeks averages, and 6 months averages, and explicitly capture inter-granularity correlation and intra-granularity sequential information.

**Modeling granularity variates.** As mentioned in Section 3.1 the shared spatial encoder transforms each SIC into independent 1D tokens. These individual spatial tokens are then concatenated sequentially based on their granularity respectively and utilized to form the multi-granularity representation. As each granularity incorporates a different time span, the dimensions of concatenated granularity sequences are mismatched. Considering that both the weekly average and monthly average are derived from daily SIC values, we choose to tokenize those sequences further and align their feature dimensions with the length of daily input using a linear transformation. The generated multi-granularity variates are subsequently fed into the sequential modeling backbone. Encouraged by prior work (Liu et al., 2023), we propose to adopt an encoder-only Transformer architecture as the sequential modeling backbone for multi-granularity fusion in Figure 3 that: (1) applies multi-head self-attention on the embedded granularity variate tokens to explicitly capture inter-granularity correlations; (2) each granularity variate is independently processed by FFN to extract intra-granulairty information (as depicted in Figure 4(a)). As for the conventional usage of vanilla Transformer in sequence prediction, the attention mechanism is applied on embedded temporal tokens which comprise variate information collected from the same time step (as in Figure 4(b)). The vanilla Transformer is challenged in forecasting series with larger lookback windows due to performance degradation and computation explosion. Furthermore, the temporal token embeddings incorporate multiple variates that represent distinct physical measurements, which may struggle to capture variate-specific representations and potentially lead to the generation of incoherent attention maps. However, in sea ice modeling, each dimension of the tokenized granularity variate incorporates SIC features that come from a different time span. This could lead to poor representation of sequential SIC features and restrict the effective modeling of inter-granularity correlations. Experimentally, we will show in Section 4.4 that by adopting our sequential modeling, the overall performance is superior to alternative backbones. After each SIC granularity variate token is properly fused and encoded, the final prediction of future granularity variate features is generated through a linear projection layer.

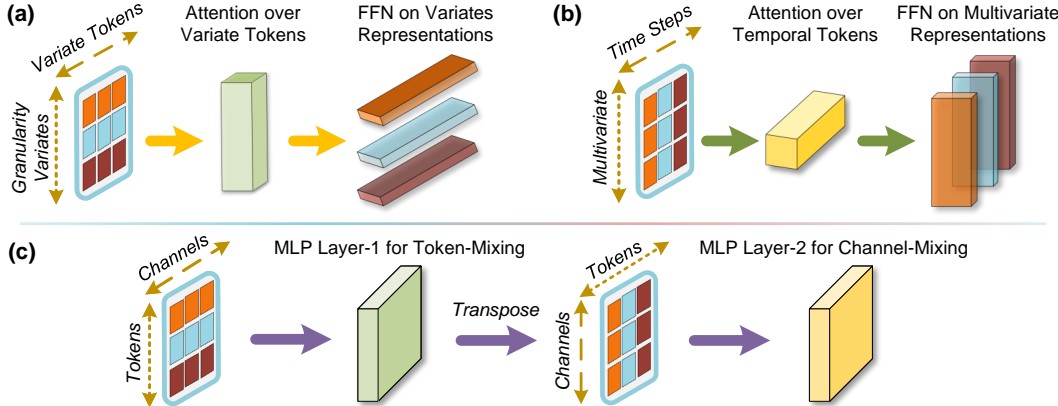

Figure 4: **Comparison between different backbones for temporal sequence modeling**: **(a)** Our proposed SIFM sequentially concatenates independent SIC tokens that are derived from each temporal scale as a granularity variate and applies an attention mechanism over the embedded variate tokens. The FFN transforms the variate representation for the input of the next layer; **(b)** For vanilla Transformer architecture (Vaswani, 2017), it applies an attention mechanism over temporal tokens and FFN is applied on multivariate representations; **(c)** The MLP-mixer (Tolstikhin et al., 2021) approach first performs token-wise mixing, then transpose the extracted features to apply channel-wise mixing. The vanilla Transformer and MLP-mixer both fall short of modeling the sequential information of sea ice.

**Sequential feature skip connection.** Considering the concatenated sequence of SIC features are linearly transformed and aligned to form the multi-granularity variate representation, the significant original sequential feature needs appropriate preservation. Besides, the deep sequence encoding process could introduce unintended noise that deteriorates the modeling of intra-granularity correlation. To compensate for the intra-granularity information and reduce the potential impact that impairs inter-granularity modeling, we propose to utilize the cross-attention mechanism as a sequential skip connection (as in Figure 3), where the latent query features are sourced from the concatenated sequence token before the linear projection and the predicted SIC sequence generates both key and value latent representations. The details about this process can be found in Appendix A.1.

## 4 EXPERIMENTS

In this section, we evaluate the forecasting performance of our SIFM over 8 years of SIC data and compare it with other deep learning models. The implementation details of our SIFM is provided in Appendix A.2.

### 4.1 DATASETS

We evaluate our proposed SIFM framework on the G02202 Version 4 dataset from the National Snow and Ice Data Center (NSIDC). It records daily SIC data starting from October $25^{th}$ 1978 and provides the coverage of the pan-Arctic region (N:-39.36°, S:-89.84°, E:180°, W:-180°). Each daily SIC data is formed of 448 x 304 pixels and each pixel corresponds to the area of a 25km x 25km grid. The SIC data has a range of 0% to 100% and areas where SIC value is greater than 15% indicate the SIE. We choose data from October $25^{th}$ 1978 to the end of 2013 as the training dataset, the years 2014 and 2015 are selected as validation set, and data collected from 2016 to 2023 are used to test models.

### 4.2 EVALUATION METRICS

To evaluate SIFM, we select widely used root mean square error (RMSE) and mean absolute error (MAE) for comparison of forecasting accuracy. We also leverage $R^2$ score to evaluate the perfor-

mance:

$$R^2 = 1 - \frac{RSS}{TSS}. \tag{2}$$

where RSS represents the sum of squares of residuals and TSS denotes the total sum of squares. The Integrated Ice-Edge Error score (Goessling et al., 2016) is introduced to evaluate the prediction of SIE:

$$IIEE = O + U, \tag{3}$$

$$O = \sum (Max(SIE_p - SIE_t, 0)), \tag{4}$$

$$U = \sum (Max(SIE_t - SIE_p, 0)), \tag{5}$$

$$SIE_p, SIE_t = \begin{cases} 1, SIC > 15 \\ 0, SIC \leq 15 \end{cases} \tag{6}$$

where O and U represent the overestimated and underestimated SIE between the prediction ($SIE_p$) and the ground truth ($SIE_t$), respectively. The difference between the forecasted and ground truth sea ice area (in millions of $km^2$) is calculated as follows:

$$SIE_{dif} = \frac{\sum(|SIE_p - SIE_t|) \times 25 \times 25}{1000000}. \tag{7}$$

We also adopt the Nash-Sutcliffe Efficiency (Nash & Sutcliffe, 1970) to further evaluate the predicted quality:

$$NSE = \frac{1 - \sum((SIC_t - SIC_p)^2)}{\sum((SIC_t - Mean(SIC_t))^2)} \tag{8}$$

### 4.3 MULTI-GRANULARITY FORECASTING

**Baselines.** Since our SIFM simultaneously generates predictions of three granularities, we select corresponding forecasting deep learning-based models for comparison. Specifically, we re-implemented SICNet (Ren et al., 2022) and trained under an identical environment for direct comparison on 7 days SIC forecasting; Due to dataset and code accessibility, we adopt performance of sub-seasonal forecasting methods as SICNet$_{90}$ (Ren & Li, 2023), IceFormer (Zheng et al., 2024), and seasonal forecasting methods IceNet (Andersson et al., 2021), MT-IceNet (Ali & Wang, 2022) that reported in the original paper for reference.

Table 1: **Quantitative results of SIC forecasting**. We compare the performance of SIFM in each temporal granularity with corresponding deep learning based methods. * marks that the performance figures are reported in their original papers for reference.

| Temporal Scale | Lead Times | Methods | RMSE↓ | MAE↓ | $R^2$↑ | NSE↑ | IIEE↓ | $SIE_{dif}$↓ |
|---|---|---|---|---|---|---|---|---|
| Sub-seasonal | 7 Days (Daily) | SICNet | 0.0490 | 0.0100 | 0.982 | 0.979 | 976 | 0.0718 |
| | | **SIFM** | **0.0429** | **0.0096** | **0.987** | **0.985** | **926** | **0.0380** |
| | 45 Days (Daily) | IceFormer* | 0.0660 | 0.0201 | 0.960 | - | - | - |
| | 90 Days (Daily) | SICNet$_{90}$* | - | 0.0512 | - | - | - | - |
| | 8 Weeks Average (Weekly) | **SIFM** | **0.0625** | **0.0140** | **0.973** | **0.968** | **1600** | **0.1541** |
| Seasonal | 6 Months Average (Monthly) | IceNet* | 0.1820 | 0.0916 | 0.567 | - | - | - |
| | | MT-IceNet* | 0.0777 | 0.0197 | 0.915 | - | - | - |
| | | **SIFM** | **0.0692** | **0.0166** | **0.917** | **0.910** | **2156** | **0.2083** |

**Main results.** The overall performance of SIFM and baseline methods are listed in Table 1. The lower RMSE/MAE indicates a more accurate forecast in SIC values. Methods with lower IIEE/SIE$_{dif}$ are more capable of identify the edge of sea ice while higher $R^2$/NSE suggests that the predicted spatial patterns are more close to the ground truth. Our proposed method achieves the best performance in all metrics for forecasting 7 days SIC, establishes a new state-of-the-art method for sub-seasonal weekly average prediction, and presents superior seasonal SIC forecasting capability. Considering the fact that baseline methods, except for SICNet, utilizes several additional atmospheric and oceanic variables to facilitate forecasting, and our SIFM only leverages SIC data with carefully extracted intrinsic inter-granularity correlation, it verifies the effectiveness of the proposed approach for multi-granularity forecasting.

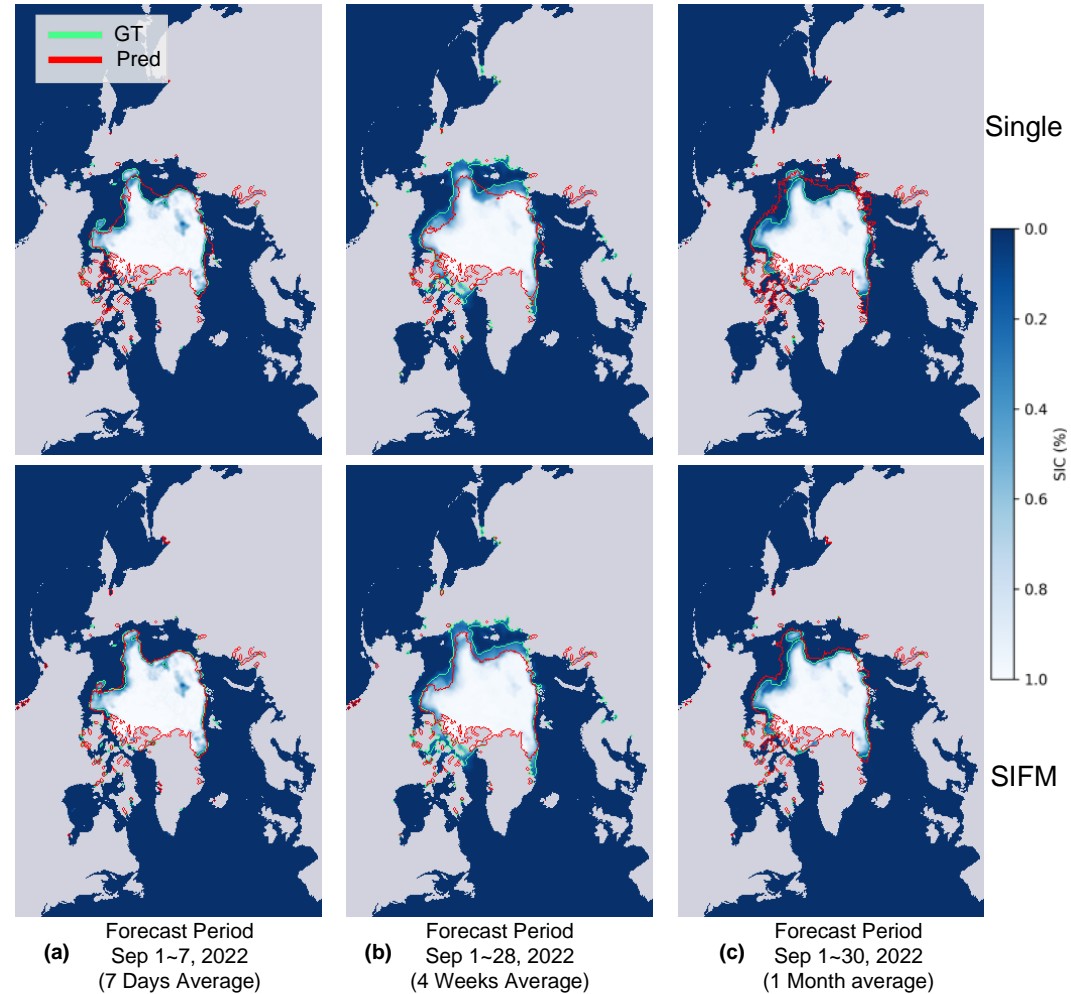

Figure 5: **Qualitative analysis of SIE prediction.** The derived SIE ground truth and prediction generated by SIFM and three single-granularity models (one for each temproal granularity) over: **(a)** The first week of September; **(b)** 4 weeks; **(c)** 1 month. Considering the abnormal increase of Arctic sea ice in 2022, our proposed method could still produce reasonable forecasts that keep the similar overall shape of Arctic SIE.

**Qualitative Analysis.** To visually verify the forecasting skills of SIFM, we select the end of the melting season in September 2022. From Figure 1(a) we can observe that the annual Arctic sea ice in 2022 has increased by a considerable margin which is against the persisting long-term reclining trend. This unusual rise makes SIC and SIE difficult for our model to predict since it only learns from the data collected before 2014. Starting from September $1^{st}$, we calculate averaged SIC of 7 days, 4 weeks and 1 month that correspond to three temporal granularities of SIFM. The ground truth of calculated average SIC along with the ground truth and predicted SIE are visualized in Figure 5. The forecasting results in the lower row are produced by SIFM and the upper row represents predictions generated by three variants of SIFM that only leverage single-granularity SIC, we will discuss later in Section 4.4.

Despite the inconsistent annual trend of Arctic SIC in 2022, our method could still generate forecasts that are consistent with the average SIE in the first week of September (Figure 5(a)), and the general shape in both 4 weeks' average (Figure 5(b)) and 1-month average (Figure 5(c)). Comparing to models with similar backbone of SIFM but only leverage single-granularity SIC, the prediction of SIE are noticeably different to the ground truth indicating that SIFM could effectively leverage multi-granularity SIC to improve forecasting skills.

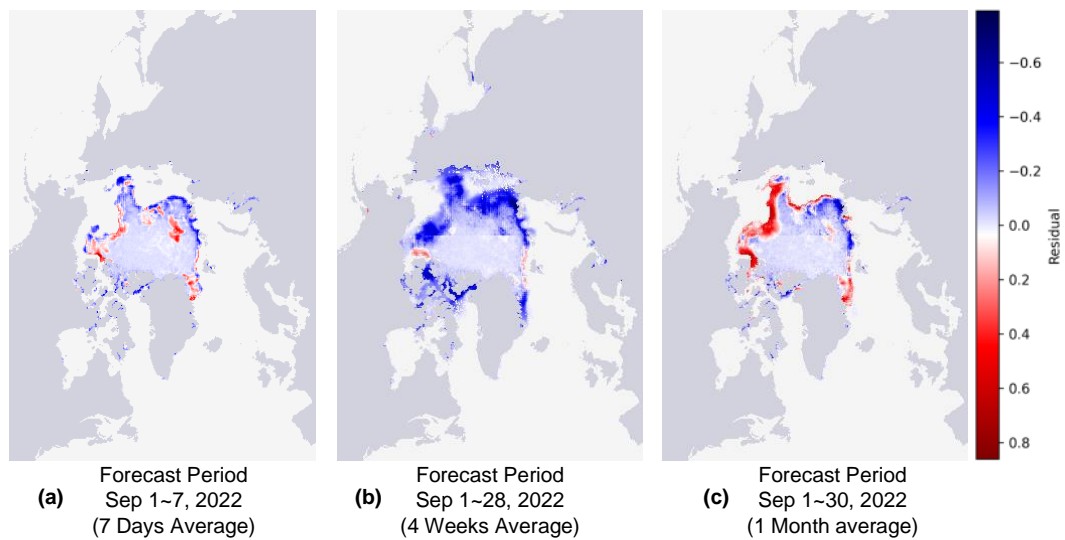

Figure 6: **Spatial residual of predicted SIC.** We examine the spatial patterns of forecasting results over the same period presented in Figure 5: SIFM could generate consistent daily forecasts **(a)**. Considering the abnormal Arctic SIC change in 2022, the annual trend could be different than the SIC data on which the model was trained, SIFM could still predict weekly **(b)** and monthly **(c)** average SIC with a bounded residual region rather than scattered forecasting results. The spatial residual is calculated by using predicted SIC to subtract the ground truth value.

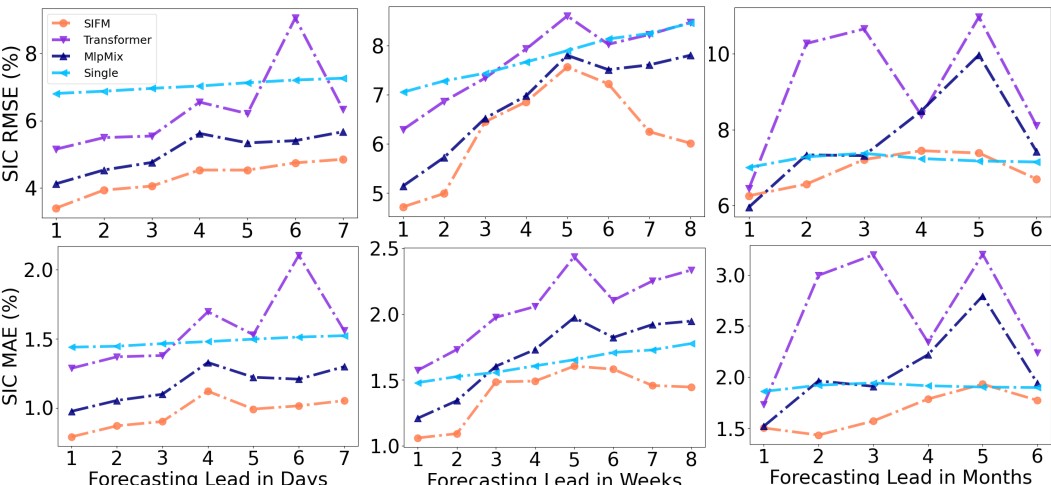

Figure 7: **Averaged intra-granularity forecasting error.** We evaluate models trained on multi-granularity and single-granularity SIC and plot RMSE and MAE of each lead time step in three temporal granularities over the test dataset.

We plot spatial residuals to further investigate the learned spatial patterns of our SIFM. In Figure 6(a), SIFM could accurately predict the first week of SIC, while in coarser weekly average granularity our SIFM tends to slightly underestimate in Arctic sea ice edge areas (Figure 6(b)). For the predicted monthly average of September 2022, the overall shape of SIE resembles the observation but overestimates SIC along the boundary.

### 4.4 ABLATION STUDY

To further analyze the performance of our proposed method, we trained five additional variants of SIFM (as in Figure 7), i.e., three single-granularity models that respectively utilize temporal

Table 2: **Effectiveness of multi-granularity representation**. *Multi* represents the proposed SIFM and *Single* stands for models with similar backbone but trained solely on single-granularity data.

| Temporal Scale | Lead Time | Granularity | RMSE↓ | MAE↓ | $R^2$↑ | NSE↑ | IIEE↓ | SIE$_{dif}$↓ |
|---|---|---|---|---|---|---|---|---|
| Sub-seasonal | 7 Days | Single | 0.0704 | 0.0148 | 0.982 | 0.979 | 1018 | 0.0509 |
| | | Multi | **0.0429** | **0.0096** | **0.987** | **0.985** | **926** | **0.0380** |
| | 8 Weeks Average | Single | 0.0771 | 0.0163 | 0.962 | 0.954 | 2208 | 0.3301 |
| | | Multi | **0.0625** | **0.0140** | **0.973** | **0.968** | **1600** | **0.1541** |
| Seasonal | 6 Months Average | Single | 0.0721 | 0.0191 | 0.882 | 0.873 | 2482 | 0.4298 |
| | | Multi | **0.0692** | **0.0166** | **0.917** | **0.910** | **2156** | **0.2083** |

Table 3: **Effectiveness of proposed approach for multi-granularity fusion**. We adopt conventional utilization of Transformer and recent trend in leveraging full MLP-based backbone (Tolstikhin et al., 2021) for temporal sequence modeling as counterparts of our proposed sequential backbone.

| Temporal Scale | Lead Time | Method | RMSE↓ | MAE↓ | $R^2$↑ | NSE↑ | IIEE↓ | SIE$_{dif}$↓ |
|---|---|---|---|---|---|---|---|---|
| Sub-seasonal | 7 Days | MLP Mixing | 0.0506 | 0.0117 | 0.984 | 0.981 | 1153 | 0.1265 |
| | | Transformer | 0.0633 | 0.0159 | 0.970 | 0.965 | 1519 | 0.2338 |
| | | SIFM | **0.0429** | **0.0096** | **0.987** | **0.985** | **926** | **0.0380** |
| | 8 Weeks Average | MLP Mixing | 0.0689 | 0.0169 | 0.969 | 0.963 | 2222 | 0.3839 |
| | | Transformer | 0.0771 | 0.0206 | 0.970 | 0.964 | 1718 | 0.2028 |
| | | SIFM | **0.0625** | **0.0140** | **0.973** | **0.968** | **1600** | **0.1541** |
| Seasonal | 6 Months Average | MLP Mixing | 0.0775 | 0.0206 | 0.857 | 0.845 | 2477 | 0.3837 |
| | | Transformer | 0.0913 | 0.262 | 0.833 | 0.821 | 3490 | 0.4902 |
| | | SIFM | **0.0692** | **0.0166** | **0.917** | **0.910** | **2156** | **0.2083** |

granularities in SIFM, and two multi-granularity forecasting models with different backbones to perform the multi-granularity fusion.

**Effectiveness of Multi-granularity modeling.** We first verify our proposed multi-granularity modeling approach by comparing SIFM with models that comprise of similar model architecture but only adopt single granularity SIC data. Comprehensive experiments in Table 2 show that by leveraging the naturally derived multi-granularity SIC, the overall performance in all temporal scales can be promoted by a significant margin. For each individual forecasting lead time, SIFM consistently outperforms models solely trained on single-granularity data (as shown in Figure 7).

**Alternative backbone for multi-granularity fusion.** To validate the effectiveness of our proposed multi-granularities fusion and sequential modeling approach, we compare the performance of our SIFM with two other variants that are trained on the identical multi-granularity data with different sequential backbones, i.e., Transformer and MLP mixer (Tolstikhin et al., 2021; Ekambaram et al., 2023). Considering intra-granularity performance in Figure 7, SIFM presents superior forecasting skill in each time step of daily, weekly average and monthly average when compared to multi-granularity variants, indicating the effectiveness of multi-granularity SIC variates for sequential modeling. As shown in Table 3, our SIFM outperforms these variants by a great margin, demonstrating the intra-granularity and inter-granularity correlations inherent in the sea ice modeling benefits for the forecasting.

## 5 CONCLUSION AND FUTURE WORK

In this paper, we propose SFIM, a transformer-based sea ice foundation model that unifies multi-granularity covering from sub-seasonal to seasonal scale to enhance the sea ice concentration forecasting. Specifically, we propose to explore the independent spatial tokens representation of SIC to exploit the inter-granularity information. These spatial tokens will be concatenated within their own granularity and go through multi-granularity fusion to explicitly model their inter-granularity correlations. Experiments demonstrate that our SIFM fulfills skillful forecasting in each granularity compared with single-granularity methods. Since our SIFM is a versatile framework, the multi-granularity climate information could also be incorporated easily in future work.

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

# A  APPENDIX

## A.1  THE DETAILS OF SEQUENTIAL FEATURE SKIP CONNECTION

$$CrossAttention(Q, K, V) = softmax(\frac{QK^T}{\sqrt{d}}) \cdot V,$$

$$Q = W_Q^g \cdot z_{in}^g, K = W_K^g \cdot z_{pred}^g, V = W_V^g \cdot z_{pred}^g \tag{9}$$

where $g$ denotes each granularity. $z_{in}^g, z_{pred}^g \in \mathbb{R}^{1 \times d_z}$ represents the sequential features before linear projection and the prediction, respectively. $W_Q^g, W_K^g, W_V^g \in \mathbb{R}^{d \times d_z}$ are the query, key and value projection matrices.

## A.2  IMPLEMENTATION DETAILS

We first generate SIC data for three granularities and trained our SIFM on this prepared dataset for 20 epochs. For each temporal granularity, SIFM outputs the same length of input data as forecasting leads, i.e. 7 days, 8 weeks average, and 6 months averages. In this study, we utilize a sliding window with a length of 30 days to generate averaged SIC in monthly granularity. The dimensions of embedding features of shared spatial encoder and decoder are set to 32. They both comprise of 4 layers of Swin Transformer V2 blocks, specifically, the number of blocks for each layer is configured as $[2, 2, 6, 2]$. We use a window size of 28 by 19 to be consistent to the ratio of SIC data.

The dimension of each individual SIC token generated by linear projection equals 128. The sequentially concatenated SIC tokens are further aligned and transformed to multi-granularity representation which has a dimension of 3 by 896. The embedding dimension of multi-granularity fusion backbone is set to 256. SIFM is trained by AdamW using Pytorch on four NVIDIA A100 80GB GPU for all experiments.

## A.3  VISUALIZATION OF FORECASTING RESULTS

In this section, we will present more visualization of forecasting results generated by SIFM.

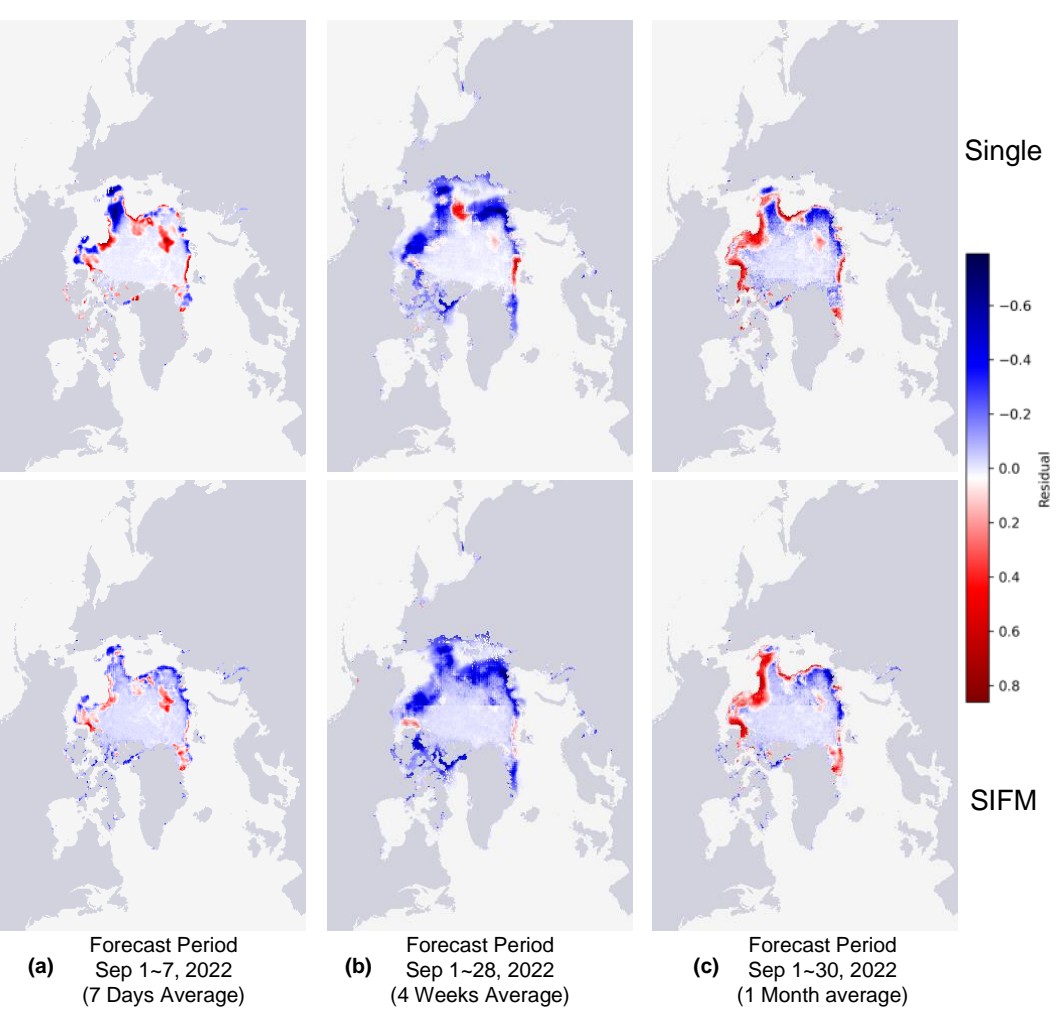

Figure 8: **Spatial residual comparison.** We compare the spatial patterns of forecasting results produced by SIFM and single-granularity variants.

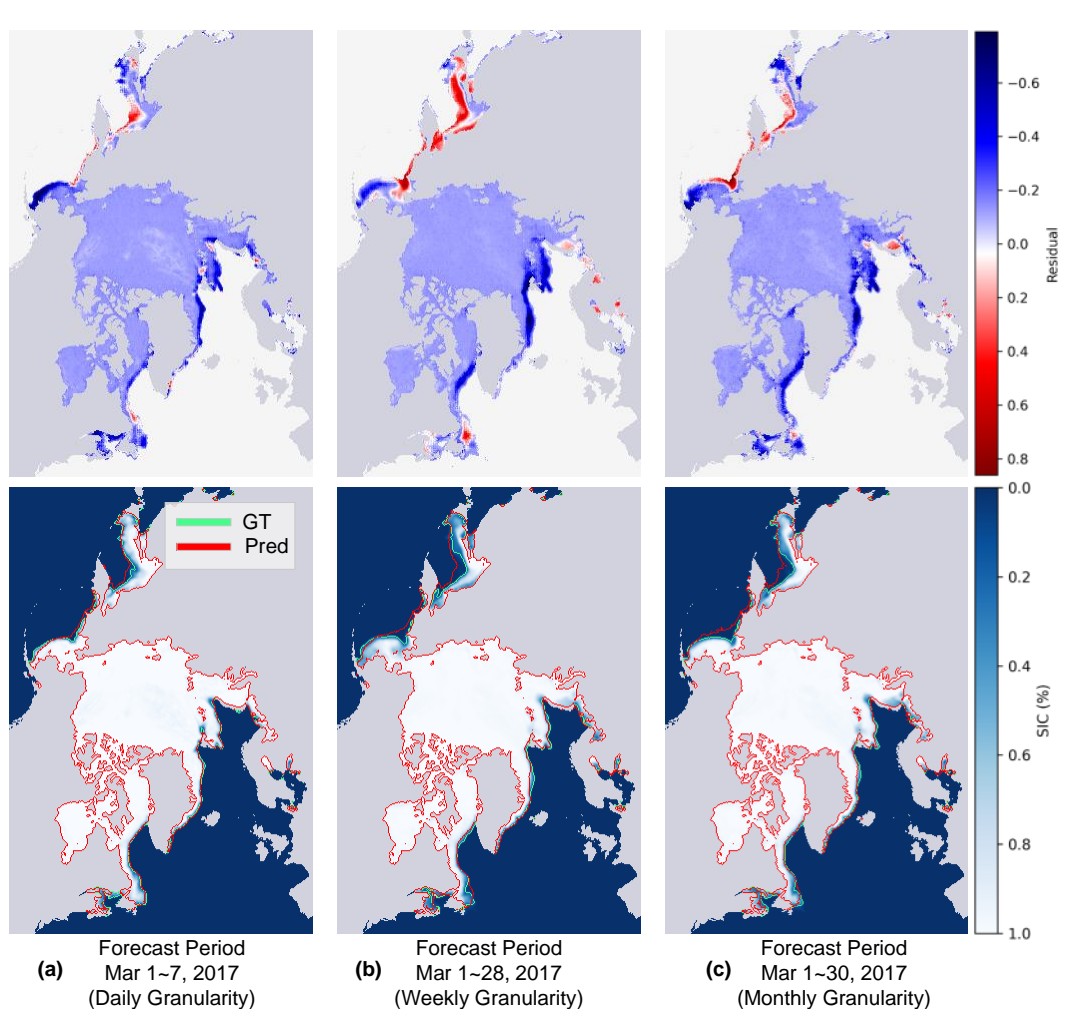

Figure 9: **Spatial residual and predicted SIE quality of Mar 2017.**

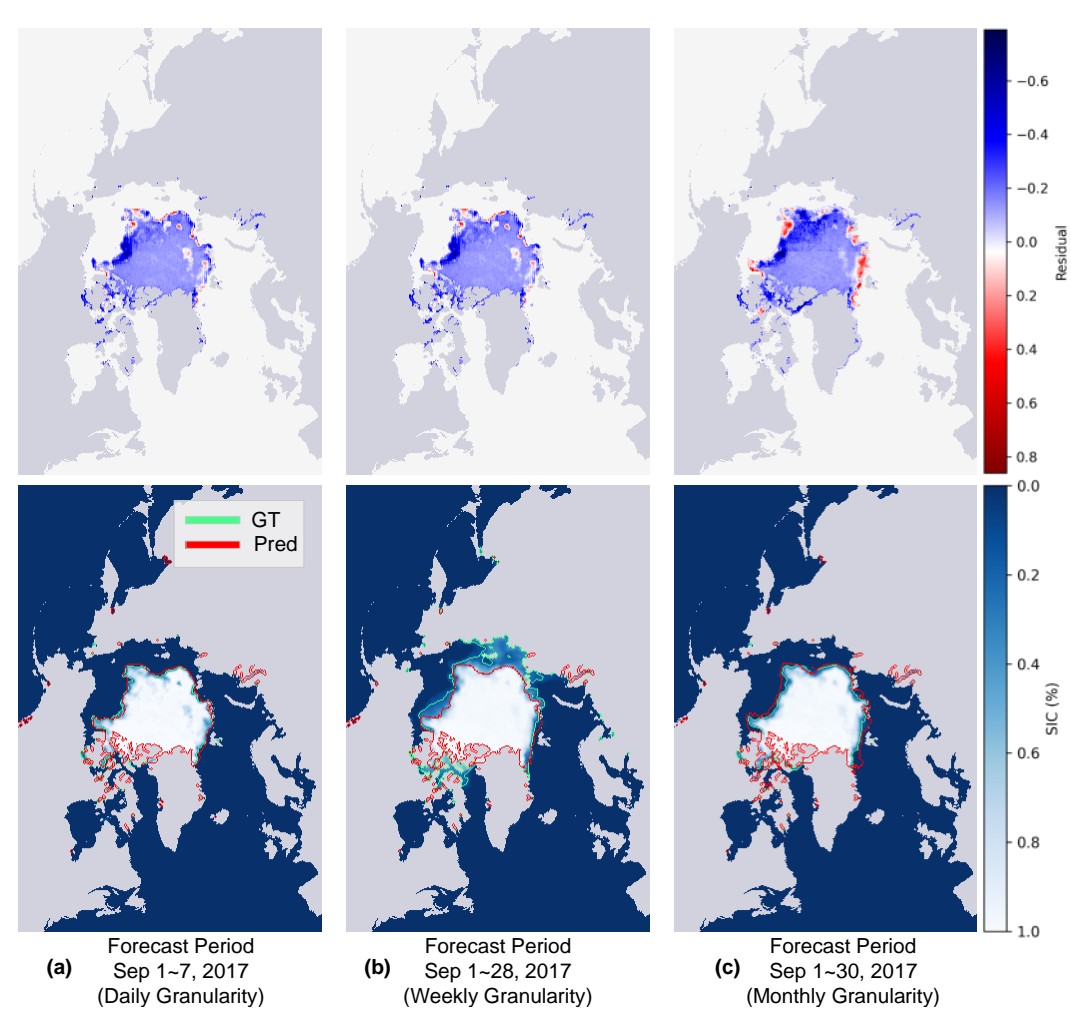

Figure 10: **Spatial residual and predicted SIE quality of Sep 2017.**

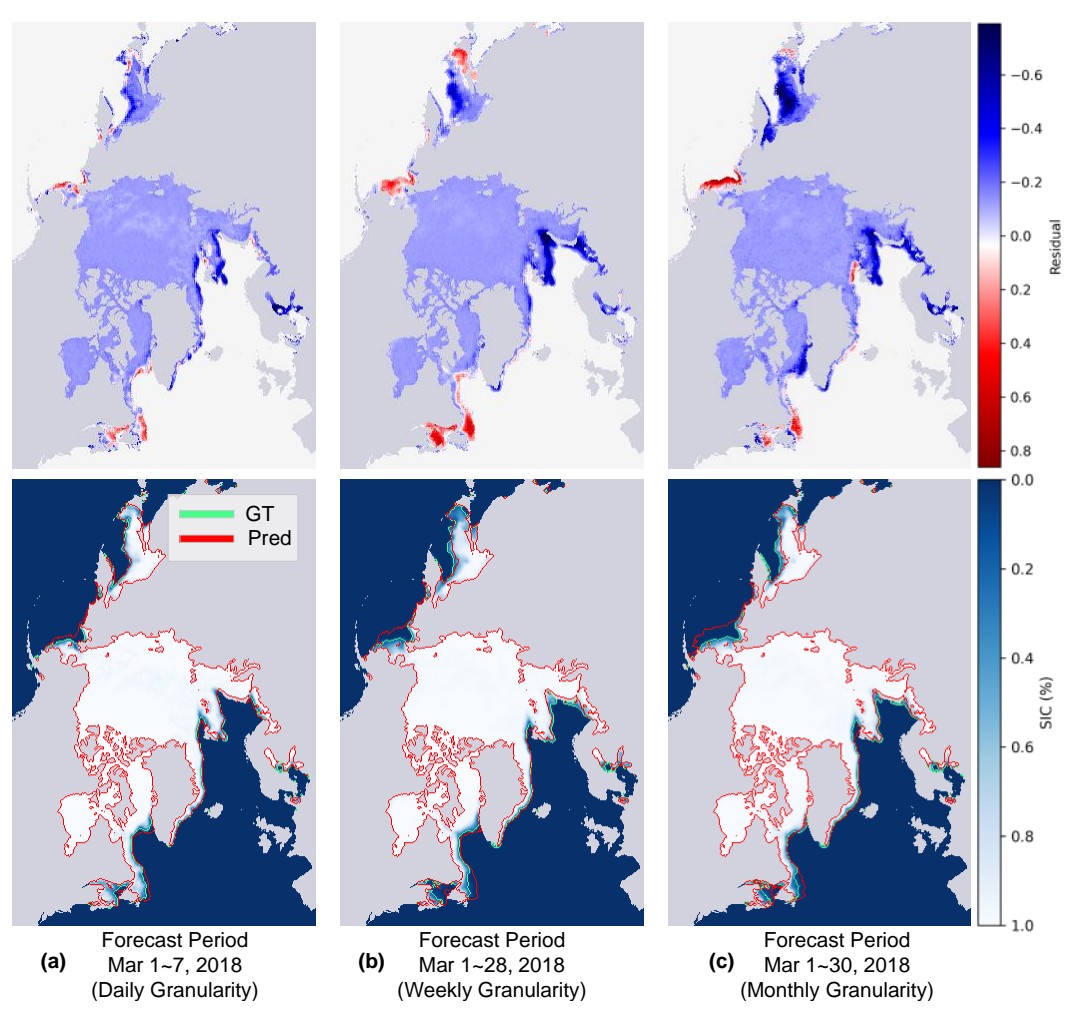

Figure 11: **Spatial residual and predicted SIE quality of Mar 2018.**

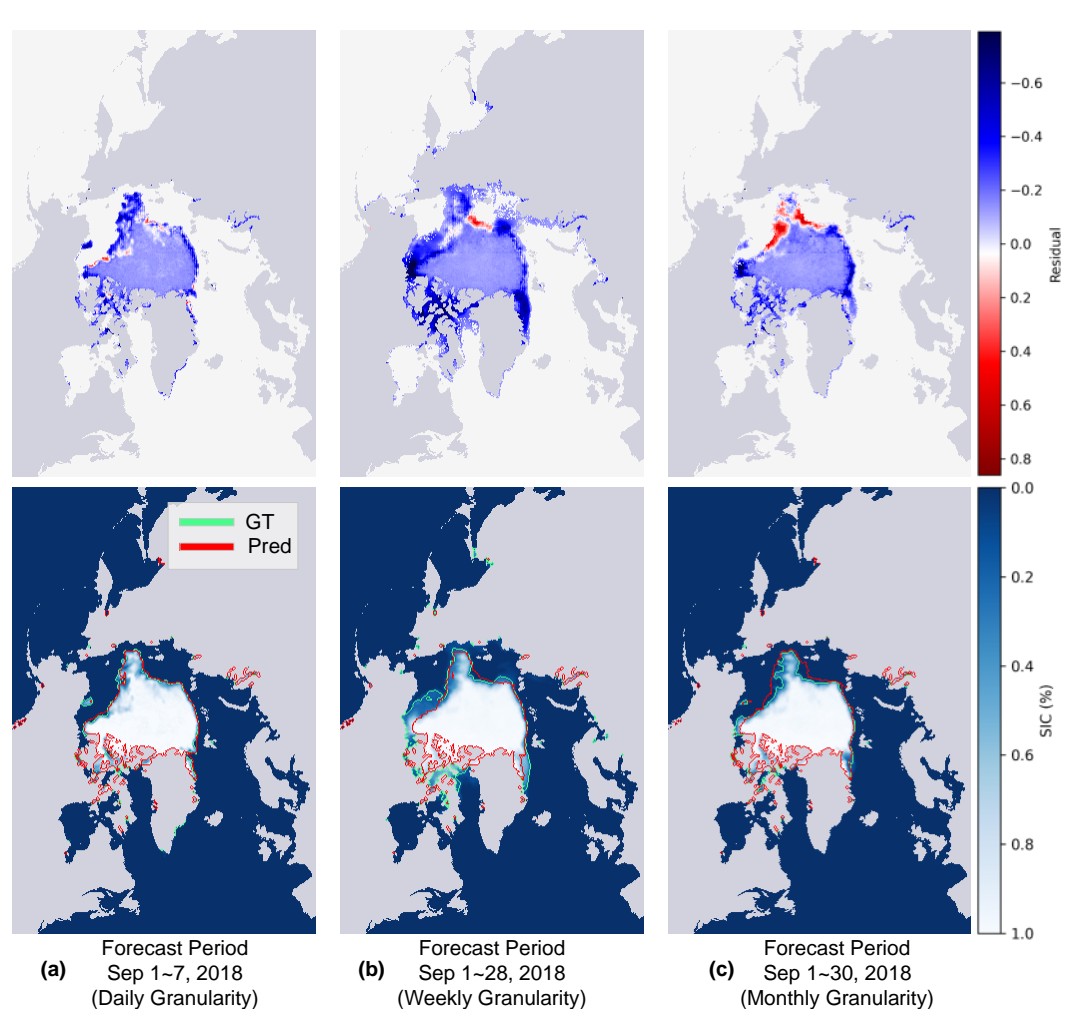

Figure 12: **Spatial residual and predicted SIE quality of Sep 2018.**

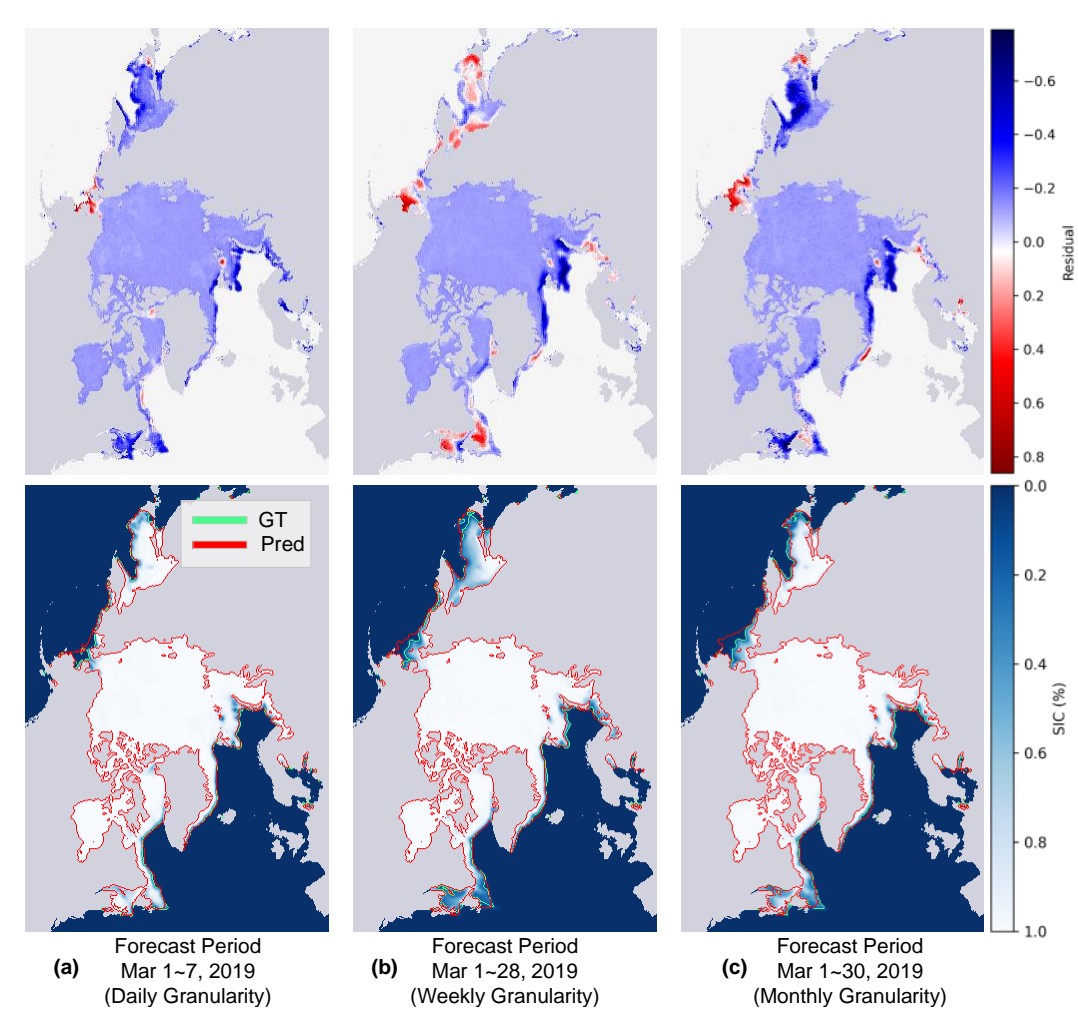

Figure 13: **Spatial residual and predicted SIE quality of Mar 2019.**

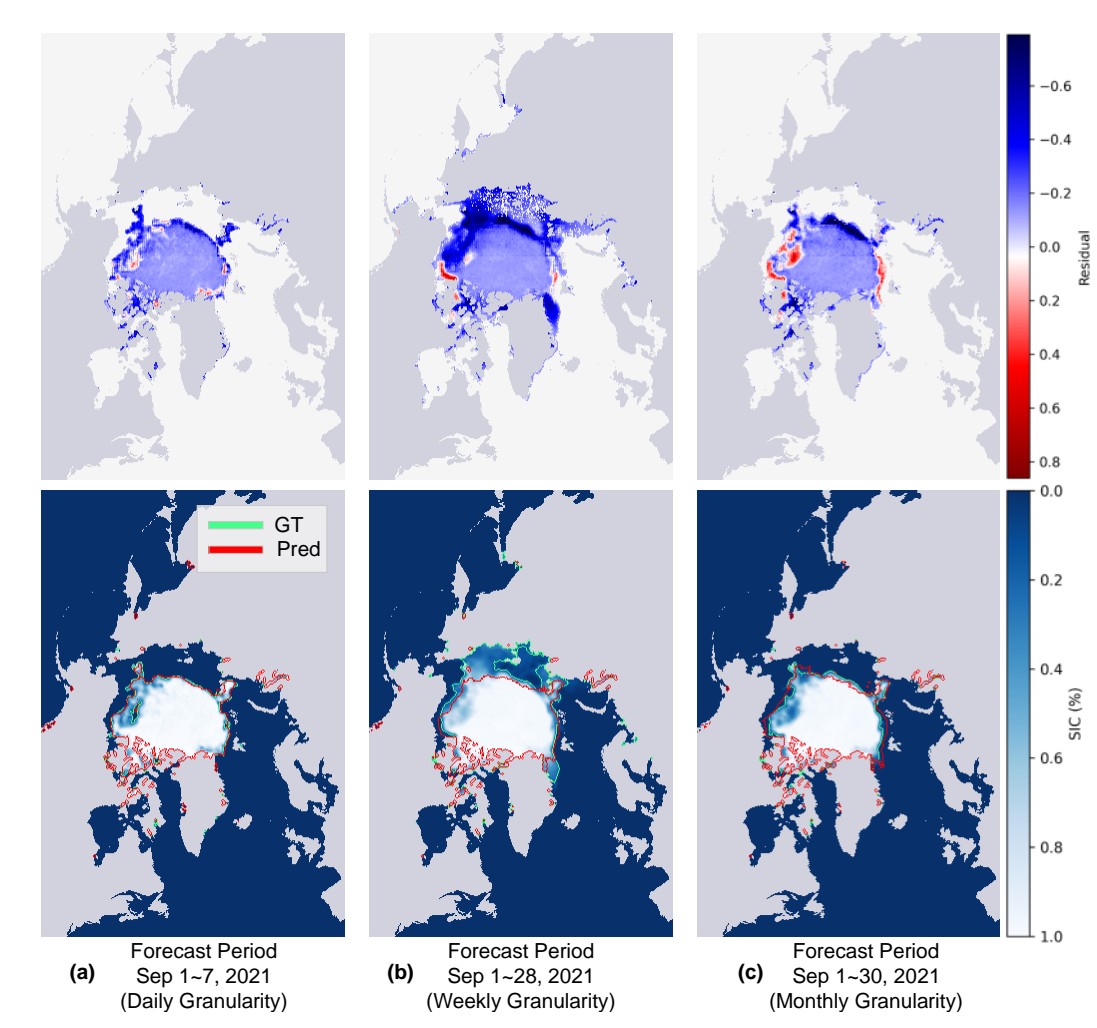

Figure 14: **Spatial residual and predicted SIE quality of Sep 2021.**

