# OpenReview forum: "SIFM:  A Foundation Model for Multi-granularity Arctic Sea Ice Forecasting"
_ICLR.cc/2025/Conference — ICLR 2025 Conference Withdrawn Submission_

### Official Review · Reviewer_QWnu · 2024-10-28

**Soundness:** 2
**Presentation:** 2
**Contribution:** 2
**Rating:** 3
**Confidence:** 5

**Summary:**

This paper introduces the Sea Ice Foundation Model (SIFM), a deep learning approach for forecasting Arctic sea ice concentration (SIC) by leveraging multi-granularity temporal data. Unlike existing methods that predict SIC at fixed temporal scales, SIFM integrates information across short-term, sub-seasonal, and seasonal granularities, capturing crucial cumulative and inter-granular effects. By using Arctic sea ice reanalysis data, SIFM learns consistent representations that enhance forecasting accuracy. Experimental results demonstrate SIFM’s superior performance over traditional models restricted to single temporal granularities, making it a valuable model for more precise SIC predictions that benefit climate studies and Arctic ecosystem monitoring.

**Strengths:**

The investigated problem is novel and interesting.

**Weaknesses:**

The sea ice forecasting problem is indeed a spatial-temporal forecasting task within a widely-researched area, yet the current study does not include a comparison with established spatial-temporal forecasting methods. Notably, the incorporation of multi-scale temporal data (daily and weekly) has been widely studied and should be properly cited to situate this work within prior research.

The contributions outlined in the manuscript are not yet robust enough for a top-tier submission. The first and second contributions essentially address similar aspects and should be consolidated into a single point, leaving only two clear and distinct contributions overall.

The concept of "independent spatial tokenization" requires further clarification. Specifically, what distinguishes this from existing spatial processing techniques needs to be articulated. The authors claim in line 123 that previous models utilize 2D convolutions to extract spatial features, which could also be seen as a form of spatial tokenization since patches are independently processed using a shared kernel. This raises the question of what makes the current approach distinct.

The naming conventions and visual representations between Figure 2(a) and Figure 2(b) should be unified for consistency. In Figure 2(a), "channel-wise fusion" is implemented with a 2D CNN and functions as the spatial encoder, aligning it with the spatial encoder in Figure 2(b). Additionally, while U-Net is an approach for modeling both spatial and sequential correlations, the alternative is presented conceptually as the "sequential modeling" and "spatial decoder" in Figure 2(b). This lack of consistent terminology leads to confusion regarding the model architecture.

The chosen baselines are not representative of recent advancements in time series forecasting. Several are outdated (published in 2021) or lack competitiveness, as indicated by their publication in less recognized venues. Incorporating more recent and widely recognized methods would strengthen the comparative analysis.

**Questions:**

Please refer to the weaknesses.

---

### Official Review · Reviewer_cm5E · 2024-11-01

**Soundness:** 2
**Presentation:** 2
**Contribution:** 2
**Rating:** 3
**Confidence:** 4

**Summary:**

This paper presents the Sea Ice Foundation Model (SIFM), a transformer-based approach aimed at forecasting Arctic sea ice concentration (SIC) across multiple temporal granularities, from sub-seasonal to seasonal. The model's goal is to capture both intra-granularity and inter-granularity information, potentially improving the consistency of SIC forecasts by considering correlations across different time scales. The authors argue that existing models, which are fixed to specific granularities, miss out on such cross-scale information.

However, the paper lacks a compelling justification for why multi-granularity modeling would significantly enhance performance over current single-granularity approaches, especially given that SIC fluctuations across these timescales might not always yield useful cross-correlations. Furthermore, the methodology appears to rely heavily on token-based transformer operations, but without robust evidence of how this tokenization and fusion technique meaningfully captures unique SIC-specific dynamics. While the authors report some improvement over other models, it remains unclear if this approach genuinely advances forecasting accuracy in practical, real-world settings.

**Strengths:**

1. The authors believe that enabling a model to perform multi-granularity forecasting is important and have focused on training such a model. In the field of spatiotemporal forecasting, having a foundation model capable of making hourly, daily, and weekly predictions would indeed be valuable.

2. Compared to models in the SIC forecasting domain, the experimental results appear to show some improvement.

**Weaknesses:**

1. The authors focus on the SIC forecasting problem, but it is unclear whether this model is specifically tailored to SIC forecasting. It seems that the model could be applied to any spatiotemporal data with multi-granularity forecasting needs. Therefore, using only a single specialized dataset is insufficient to validate the model’s effectiveness. The authors should consider comparisons with mainstream video prediction backbones and datasets, such as SimVP: "SimVP: Simpler yet Better Video Prediction." CVPR 2022.

2. The methodology appears to rely heavily on token-based transformer operations, yet lacks strong evidence that the tokenization and fusion techniques effectively capture SIC-specific dynamics. Additionally, the structure of this paper lacks a solid theoretical foundation.

**Questions:**

See W1. Can the authors apply their models to general spatiotemporal forecasting tasks?

---

### Official Review · Reviewer_Kbqx · 2024-11-02

**Soundness:** 2
**Presentation:** 2
**Contribution:** 2
**Rating:** 3
**Confidence:** 5

**Summary:**

This paper presents the Sea Ice Foundation Model (SIFM), a new deep learning approach for forecasting Arctic sea ice concentration (SIC). It addresses the limitations of existing methods that operate at fixed temporal granularities, which overlook important inter-granularity correlations. SIFM simultaneously models SIC at daily, weekly, and monthly scales by independently tokenizing spatial features and utilizing an attention mechanism to capture relationships across these scales. The experimental results indicate that SIFM outperforms existing deep learning models, highlighting its effectiveness in integrating diverse temporal information for improved SIC forecasting. This research contributes a valuable perspective on enhancing predictive accuracy in Arctic sea ice studies.

**Strengths:**

1.The paper explicitly models sea ice concentration prediction at multiple temporal scales, which enhances the understanding of temporal dynamics in forecasting.

2.This paper consistently outperforms existing deep learning approaches across various metrics, demonstrating its capability to better capture sea ice dynamics.

3.Extensive ablation experiments validate the contributions of different model components, reinforcing the effectiveness of the multi-granularity approach.

**Weaknesses:**

1.The idea of the multi-granularity approach is commendable. However, it is not a ‘Foundation Model’ for SIC forecasting. A foundation model generally refers to a large parameter scale model trained on large-scale data, which is an overclaim about its effectiveness.

2.The model treats the SIC forecasting problem within a general spatio-temporal analysis framework without addressing the unique characteristics of sea ice prediction. There is limited exploration of the specific oceanographic and climate science features that influence these predictions.

3.After the spatial feature encoding, the model applies linear mapping to weekly and monthly features for the alignment of multi-granularity tokens. This approach may disconnect the temporal context of these tokens, raising concerns about whether time series information is preserved and how different granularity tokens are interpreted within the attention mechanism.

4.Although the paper includes thorough ablation studies, the comparative experiments are insufficient. The authors use many performance metrics from the original papers without providing comparisons based on the new metrics introduced. Some methods are not evaluated within a unified framework, particularly regarding daily versus varying time frames (e.g., 7, 45, 90 days), and there is a lack of comparisons at the weekly granularity.

5.There exists some typos confusing the understanding. For example, the abstract contains a misstatement regarding "inter-granularity," which should be "intra-granularity."

6.The direct flattening of features obtained from the spatial encoder raises questions about potential loss of spatial correlation information in vertical dimension, as neighboring features may lose their adjacency relationships after flattening.

**Questions:**

Please address my concerns in the weakness part.

---

### Official Review · Reviewer_kPm7 · 2024-11-03

**Soundness:** 3
**Presentation:** 3
**Contribution:** 3
**Rating:** 6
**Confidence:** 4

**Summary:**

The paper proposes to combine different temporal aggregations of image time series (daily, weekly and 6 months averages, called granularities) as inputs for a transformer architecture, in order to model Sea Ice Concentration (SIC). The paper proposes to forecast the SIE at all the granularities as well. The argument is that these different granularities will inform the model of the different short-, and medium-term temporal dynamics (sub-seasonal and seasonal variations), without explicitly modeling them.

**Strengths:**

The paper is mostly well written and well constructed. The motivation behind the work is clear and the different concepts well explained.
The schematic figures are very well done, and help a lot in the reading of the paper.

Overall the model seems to perform very well, and outperforms the other proposed models.
The study also performed quite a bit of ablation studies, which allows to better understand the strength and weaknesses of the approach.

**Weaknesses:**

## "Foundation" model

The term foundation model feels very misplaced in this paper, and more like a term added as an afterthought because of the current trends in the deep learning community.
Generally, foundation models are models considered to be trained on a very large amount of data, and not for a particular task. The model is then being fine tuned on different down-stream tasks for particular applications. This is not the case here, as the model is not trained on multiple tasks, but rather trained to predict multiple granularities, which is one task. Additionally, the model is suited for this particular task, but could hardly be applied to something different, given the specificity of the application.

The paper reads as if, initially, the letter "F" in Sea Ice Foundation Model, was supposed to be "Fusion", but then changed last minute to follow a trend. The model is never argued as a foundation model, and save for the name, the term is never used. The concept of fusion on the other hand, is largely covered and mentioned multiple times throughout the paper.

I would strongly suggest changing Foundation to Fusion.

## Related Work

Line 137: "temporal information inherent in sea ice modeling can not be fully exploited". I would argue that LSTM combined with CNNs can actually extract the spatio-temporal information quite well. Maybe not particularly in Sea Ice modeling, but in other fields there is a slue of models making use of the spatio-temporal information, be it transformers or combined LSTM/CNN architectures (c.f. Yu et al. 2024, Giezendanner et al. 2023, Boulila et al. 2021, etc...).

## Temporality
### Terms
The temporalities are a bit all over the place. With a paper like this, where the temporal resolution is central to understanding the work, I feel like the wording is particularly important.
Line 186 explains the granularities relatively well: daily, weekly average, and monthly average. But after that, the terms get a bit muddy:
- Line 174: "7 days, 8 weeks’ averages and 6 months’ averages". My understanding is that these are the time steps. You take 7 days of daily data as input, 8 weeks of weekly average and 6 month of monthly average. Just reading the figure, this is not clear, and I had to piece this together from the text.
- Line 245: "We propose to jointly model three granularities that cover sub-seasonal to seasonal scale, i.e., 7 days, 8 weeks averages, and 6 months averages". Here the confusion is quite big. Area you taking 8 time steps of weekly average, or an average over 8 weeks? Same for month. I think I got the gist, but this is far from clear and I had to go over the paper multiple times to piece it together.
- Table 1 (line 358): Here it gets worse. Lead Times: 7 days (daily). Intuitively I read this as the 7 days lead time, temporal resolution of 1 day, i.e. the values of the 7th day. But I'm not sure this is right, since you mention you forecast at all time steps, so is the result the error of the 7th day, or the cumulative/average error over the 7 days? Same comment for 8 weeks and 6 month.

I think the paper would profit a lot from a figure describing the temporalities. Show exactly what the inputs and outputs are, and how you define the terms used in the rest of the paper.

### Number of Time steps per granularity

I might have missed it, but it is unclear to me if the number of time steps per granularity is justified anywhere.
Why take only the seven next days? Why not the 6 months of daily inputs as a whole? The multi-granularity approach probably makes the model lighter, but a model with 6 month of daily inputs could very well perform as well as the multi-granularity one.
Alternatively, one could also model the daily output, with a moving window, and average over the other other granularities.

## Dataset

It is not totally clear to me, from section 4.1, how big the dataset is. A rough estimation of the daily data from 1978 to 2013 for the training data yields ~13'000 training samples), but how were the different temporalities sampled? Is there a moving average? Does that mean the validation set has 365 points (assuming 6 months at the begining and the end have to be blocked for the monthly average)? And the test set 2,555?

## Results

Table 1: The results for the 7 days daily lead time are better, but almost seem marginal. The table is missing units, but assuming it's percentage, is a difference in RMSE of 0.0061 really significant?
The comparisons to 45 Days and 90 Days are hard without having the output of the model. Would it be possible to run the model with 45 and 90 days of daily inputs to compare?
Additionally, an ablation study on the number of time steps per granularity would be interesting.

The results on the 6 months seem more significant.

The ablation study "Effectiveness of multi-granularity representation" is interesting, but it would also be interesting to see the results for a single granularity aggregated on the other granularity. For example: daily forecast, aggregated over 6 months.

## Conclusions

The conclusion is very minimal, and would profit from a bit more of a discussion. What advancements do the results of this model bring?

## Minor comments

Line 44 and 131: "However, numerical and statistical models usually rely on high-performance computing on CPU clusters and often lead to complex debugging processes and uncertain parameterization" and "tend to result in complex debugging processes and uncertain parameterization". That might be true, nevertheless Deep learning methods can also have large debugging processes as well as uncertain parameterizations.

## Grammar comments

Line 144: move "while" to the beginning of the sentence.
Line 146: remove "in the vision community"

## Comments on figures

Line 063: "(a)" should be bold
Line 173: It would be helpful to have the numbers mentioned in the caption reflected in the figure (e.g. (1), shared spatial encoder, (2), ... (3), ...)

Figure 6: set the value of 0 to the middle, make sure it is on the white part, and set the limits of the color bar to be the same up and down.

Figure 7: use a shared x axis for each column, and a shared y axis for each line

**Questions:**

c.f. Weaknesses

---

> ### Author Response · Authors · 2024-11-16
>
> We sincerely thank you for your time and contributions on conscientiously reviewing our paper. Your insights and detailed comments would greatly help us to continue polishing our work. Considering the overall rating given from all reviewers, we decided to withdraw our submission and further revise it based on those advice.

---

### Note · Authors · 2024-11-16

I have read and agree with the venue's withdrawal policy on behalf of myself and my co-authors.